# Ontogenetic Changes in the Expression of the Lin28 Protein in the Rat Hypothalamic Tuberal Nuclei

**DOI:** 10.3390/ijms232113468

**Published:** 2022-11-03

**Authors:** Polina A. Anfimova, Lydia G. Pankrasheva, Konstantin Yu. Moiseev, Elizaveta S. Shirina, Valentina V. Porseva, Petr M. Masliukov

**Affiliations:** Department of Normal Physiology, Yaroslavl State Medical University, ul. Revoliucionnaya 5, 150000 Yaroslavl, Russia

**Keywords:** Lin 28, hypothalamus, ventromedial nucleus, insulin signaling, aging, immunohistochemistry

## Abstract

The hypothalamus is a primary regulator of homeostasis, biological rhythms and adaptation to different environment factors. It also participates in the aging regulation. The expression of neurons containing Lin28 was studied by immunohistochemistry in male rats aged 2, 6, 12, and 24 months in the tuberal region of the rat hypothalamus. We have shown for the first time the presence of Lin28-immunoreactive (IR) neurons in the ventromedial nucleus (VMH) and their absence in the dorsomedial and arcuate nuclei in all studied animals. With aging, the percentage of Lin28-IR neurons increases from 37 ± 4.7 in 2-month-old rat until 76 ± 4.6 in 6-month-old and further decreases to 41 ± 7.3 in 12-month-old rat and 28 ± 5.5 in 24-month-old rats. Many VMH Lin28-IR neurons colocalized components of insulin signaling including mTOR, Raptor, PI3K and Akt. The percentage of Lin28/Akt-IR neurons was maximal in 6-month-old and 1-year-old rats compared to 2-month-old and 2-year-old animals. The proportion of Lin28/PI3K-IR neurons significantly increased from 77 ± 1.2 in 2-month-old rat until 99 ± 0.3 in 24-month-old rats and 96–99% of Lin28-IR neurons colocalized mTOR and mTORC1 component Raptor without statistically significant differences in all studied age groups. Thus, Lin28 expresses only in the VMH neurons of the tuberal nuclei of the hypothalamus and the Lin 28 expression changes during the development together with the components of PI3K-Akt-mTOR signaling.

## 1. Introduction

The highly conserved RNA-binding protein, Lin28, is involved in many biological processes, including development, control of onset of puberty and menopause, cell reprogramming, pluripotency and metabolism. Lin28 was initially discovered in Caenorhabditis elegans as a controller of time development and it has been found to selectively repress the expression of microRNAs, including those related to the *Let-7* family [1,2]. Mammals produce two Lin28 paralogs, Lin28 (also known as Lin28a), and Lin28b. Lin28 and Lin28b are differentially localized in the cells: Lin28 is mainly found in cytoplasm, whereas Lin28b accumulates in the nucleus [3].

The Lin28/*Let-7* axis is characterized by double negative feedback in the regulation of various biological functions. The level of let-7 is directly correlated with the expression levels of Lin28/Lin28b. Lin28 binds to the terminal loops of *Let-7* precursors, leading to inhibition of processing and the induction of uridylation and degradation of the precursor. Lin28 is also a direct translational regulator: it selectively binds to a group of mRNAs and stimulates their translation [4,5].

Lin28 and *Let-7* regulate the self-renewal and differentiation of stem cells. Lin28 or Lin28b is upregulated whereas *Let-7a* is downregulated in cancer and undifferentiated human and mouse embryonic stem cells relative to normal tissues. The *Let-7* microRNA family members act as tumor suppressors by inhibiting expression of oncogenes and pluripotency factors including *K-Ras, Cyclin D1, c-Myc, Cdc34, Hmga2, E2f2 and Lin28* [6,7]. 

The Lin28/*Let-7* system is participated in the regulation of glucose metabolism and acts as a suppressor of multiple genes involved in the insulin signaling pathway [8,9,10]. Besides, *Let 7* is involved in different pathways that regulate aging and aging-related diseases, and the level of *Let-7* is elevated in aging tissues [11,12]. 

Lin28 upregulates phosphoinositide 3-kinase (PI3K)–Akt1–mammalian target of rapamycin (mTOR) signaling, which activates proliferative processes, metabolic regulation and longevity [8]. Lin28 effects on mTOR signaling are mediated, at least in part, by mTOR complex 1 (mTORC1) [13]. Raptor is a key component of mTORC1 [14]. 

Lin28 is highly expressed in the hypothalamus in contrast with peripheral tissues. Expression Lin28 in the hypothalamus is affected by the metabolic state. Lin28 overexpression in the hypothalamus induced a significant improvement in the glucose metabolism and did not influence body weight [15].

The hypothalamus is a primary controller of homeostasis, biological rhythms and adaptation to different environment factors. It also participates in the aging regulation [16,17,18]. One of the manifestations of aging is the development of metabolic syndrome. Ventromedial (VMH), dorsomedial (DMH) and arcuate (ARH) hypothalamic nuclei participate in the regulation of metabolism and energy balance [19,20]. There are some data that neurons in the tuberal hypothalamic area control not only metabolic homeostasis but also lifespan [17,18]. In this case, expression of some markers changes in the hypothalamus of aged animals, for example calcium-binding proteins and neuronal NO synthase [21,22]. 

However, data on age-related changes in Lin28 expression are unavailable in the contemporary literature. Thus, the aim of the present study was to determine the location and percentage of Lin28-immunoreactive (IR) neurons as well as neurons co-expressing Lin28 with components of PI3K-Akt-mTOR signaling in the tuberal group of hypothalamic nuclei (ARH, DMH and VMH) in rats during aging.

## 2. Results

### 2.1. Location and Percentage of Lin28-IR Neurons

We observed Lin28-IR neurons in the VMH of all studied rats. Immunoreactivity was located in the neuronal cytoplasm and proximal dendrites (Figure 1). However, we did not find any Lin28-IR neurons in the DMH and ARH. 

In the VMH, the percentage of Lin28-IR neurons significantly increased from 37 ± 4.7 in 2-month-old rat until 76 ± 4.6 in 6-month-old (*p* < 0.01) and significantly decreased later to 41 ± 7.3 in 1-year-old rat (*p* < 0.05 vs 6-month-old) and 28 ± 5.5 in 24-month-old animal (no statistically significant differences between 12- and 24-month-old, *p* > 0.05) (Figure 2).

### 2.2. Colocalization of Lin28-IR Neurons with Components of Insulin Signaling

Many VMH Lin28-IR neurons colocalized components of insulin signaling including mTOR, Raptor, PI3K and Akt1 (Figure 3). The percentage of Lin28/Akt1-IR neurons was maximal in 6-month-old (92 ± 1.0) and 12-month-old (93 ± 1.3) rats compared to 2-month-old (55 ± 0.9) and 24-month-old (82 ± 2.1) animals (Figure 3 and Figure 4). Differences between the 2-month-old rats and other age groups as well as between the 24-month-old group and the 6- or 12-month-old rats were statistically significant (*p* < 0.05).

The proportion of Lin28/PI3K-IR neurons significantly increased from 77 ± 1.2 in 2-month-old rat to 86 ± 1.0 in 6-month-old rats (*p* < 0.05) and subsequently to 96–99% in 12-month-old and 24-month-old rats (*p* < 0.01, statistically significant differences between 6-month-old and 12-month-old rats) (Figure 3 and Figure 5). Differences between 12-month-old and 24-month-old rats were not statistically significant (*p* > 0.05).

Nearly all Lin28-IR neurons (96–99%) colocalized mTOR and mTORC1 component Raptor without statistically significant differences in the all studied age groups (*p* > 0.05) (Figure 3, Figure 6 and Figure 7).

## 3. Discussion

We first studied expression of Lin28 in hypothalamic neurons located in the VMH, DMH and ARH in young, adult and aged rats. In the current study, we first observed Lin28-IR neurons only in the VMH in rats of different age groups from young until aged animals. ARH and DMH plays an important role in the metabolic regulation. Surprisingly, we did not observe Lin28-IR neurons in the above-mentioned nuclei. The percentage of Lin28-IR VMH neurons increases during the first 6 months of life and decreases in 1-year-old and aged 2-year-old rats. Low level of Lin28 in young rats was confirmed by Sangiao-Alvarellos et al. (2013), where Lin28 mRNAs displayed very high hypothalamic expression during the neonatal period but was minimal in the pubertal period [23]. 

We found that many VMH Lin28-IR neurons colocalized components of insulin signaling including mTOR, Raptor, PI3K and Akt1. However, patterns of expression of these components were different. The percentage of Lin28/Akt1-IR neurons had a peak in 6-month-old and 1-year-old rats. The proportion of Lin28/PI3K-IR neurons increased continuously and was maximal in aged 24-month-old rats. Nearly all Lin28-IR neurons colocalized mTOR and mTORC1 component Raptor without statistically significant differences in the all studied age groups from young until aged. From other studies, high levels of mTOR, Raptor and AkT were identified in the ARH and VMH [24]. From the literature data, Lin28 overexpression in the mouse VMH showed improved glucose tolerance and insulin sensitivity, while Lin28 downregulation during high fat diet was deleterious [8,15].

Lin28 upregulates components of the PI3K/AKT/mTOR pathway [8,11]. Akt phosphorylation level was increased in Lin28 VMH overexpressing mice and decreased in Lin28 knockdown mice [15]. In our previous data, the proportion of mTOR VMH neurons increased in the first six months of life and then decreased in 12-month-old and 24-month-old rats [25], similar to the pattern of changes of Lin28-IR VMH neurons, which we observed in the current work. Genetic inhibition of mTOR signaling can increase the lifespan of yeast, worms, and flies [26,27,28]. We can suggest that the downregulation of Lin28 and mTOR in VMH neurons may have a protective role during postnatal development and aging. 

Lin28 inhibits the biogenesis of *Let-7* miRNAs, which in turn repress Lin28 post-transcriptionally [11,29]. Members of the *Let-7* miRNA family are involved in the regulation of glucose metabolism and act as suppressors of many genes involved in the insulin signaling pathway [30]. In previous studies, we found that, with aging, the expression of *Let-7a* microRNA in the nuclei of the medio-basal group of hypothalamic nuclei decreases, mainly in the DMH in males [30]. However, we did not find Lin28-IR neurons in the DMH. Nevertheless, Lin28 can affect various mRNA targets without altering *Let-7* miRNA levels and Lin28-induced activation of Akt and mTOR may be not associated with changes in *Let-7* expression [15,31].

Some recent data confirm the important role of the hypothalamus in the aging regulation. Cai group proposed microinflammation in the ventromedial hypothalamus as the main driver of aging [32,33]. Aging in mammals is accompanied by obesity and changes in hypothalamic neurons [16,34]. The VMH, which control the metabolic pathways, energy balance and peripheral circadian rhythms, may also participate in the programming of aging [35].

## 4. Materials and Methods

### 4.1. Animals

All experiments were performed in accordance with the Guidelines of the Russian Ministry of Health and EU Directive 2010/63/EU for animal experiments. Protocols were approved by the Ethics Committee of the Yaroslavl State Medical University. All necessary efforts have been made to reduce the number of animals and their suffering during the experiment. Experiments were performed on 2-month-old (weight 170–190 g), 6-month-old (weight 200–250 g), 12-month-old (weight 260–280 g), and 24-month-old (weight 330–350 g) male Wistar rats (5 animals in each group, 20 total). Animals were housed in an acclimatized room under 12 h light/dark cycle and a temperature of 22–24 °C with free access to food and water.

### 4.2. Immunohistochemistry

Immunohistochemistry was performed as previously described [21,22]. Briefly, rats were anesthetized with a lethal dose of urethane (3 g/kg, i.p.) and trans-cardially perfused within 10 min with 4% paraformaldehyde in phosphate-buffered saline (PBS, 0.01 M; pH 7.4). Brains were extracted and postfixed for 2 h in the same fixative at room temperature. After cryoprotecting in a solution of 30% sucrose in PBS (pH 7.4) at 4 °C, tissue was embedded in Tissue-Tek O.C.T. Compound (Sakura Finetek Europe, the Netherlands), frozen and kept at −20 °C. Identification of VMH, DMH and ARH was carried out with the rat brain atlas [36]. The area of interest was identified by the presence of median eminence and the extension of the third ventricle (bregma from −2.4 mm to −3.2 mm). Brain tissue was sliced in the coronal plane with a cryotome (Shandon E, Thermo Fisher Scientific, Loughborough, UK). 14-μm-thick frozen sections were mounted on SuperFrost Plus (Menzel Gläser, Braunschweig, Germany) slides.

Immunostaining for Lin28 and mTOR, Raptor, PI3K, Akt1 was performed. For that purpose, sections were washed in PBS and incubated in blocking solution containing 5% normal donkey serum and 0.3% Triton X-100 in PBS for 30 min. Then sections were exposed overnight to primary antibodies: mouse anti-Lin28, 1:200 (sc-293120, Santa Cruz Biotechnology, Dallas, USA); rabbit anti- mTOR, 1:300 (PA5-34663, Invitrogen); anti-Raptor, 1:200 (a8992, ABClonal); PI3KC3-C1, 1:200 (a12295, ABClonal); Akt1, 1:200 (a17909, ABClonal, Wuhan, China) at room temperature. On the next day, sections were washed with PBS and incubated in the secondary antibodies: FITC-conjugated donkey anti-mouse IgG (Code: 715-095-150, Jackson Immunoresearch Europe, UK) CY3-conjugated donkey anti-mouse IgG (Code: 715-165-150, Jackson Immunoresearch Europe, UK) and CY3-conjugated donkey anti-rabbit IgG (Code: 711-165-152, Jackson Immunoresearch Europe, Ely, UK) 1:100, for 2 h at room temperature. In the case of single Lin28 labeling, sections were counterstained with NeuroTrace™ 500/525 Green Fluorescent Nissl Stain (NGreen) (N21480, Thermo Fisher Scientific, Eugene, USA) 1:200. NeuroTrace fluorescent Nissl stain is selective for the Nissl substance characteristic of neurons and provides more sensitivity than traditional histological dyes like toluidine blue or cresyl violet [37]. Finally, sections were washed and cover-slipped in VectaShield mounting medium (Vector Laboratories, Burlingame, USA). For the antibody specificity, controls were performed by pre-absorption of primary antibody with the recombinant Lin28 protein (ab271598, Abcam, Cambridge, UK) or omission of the primary antibody.

### 4.3. Image Acquisition

Slides were examined on Olympus BX43 fluorescence microscope (Olympus Europa, Hamburg, Germany) equipped with the following filter cubes: U-FBWA (Blue Excitation) and U-FGWA (Green Excitation). Olympus UPLFLN 20X (Olympus UPLFLN U Plan Fluorite 20X/0.50 ∞/0.17/FN 26.5) and 100X (Olympus UPLFLN U Plan Fluorite 100X/1.30 ∞/0.17/FN 26.5) objective was used. Images were captured using a FL-20 cooled CCD digital camera (20 Megapixel) and Mosaic V2.1 imaging software (Tucsen Photonics, Fuzhou, China). 

VMH, DMH and ARH from each side were analyzed individually. Neuronal profiles with a clear identified nucleus were counted. To determine the percentage of IR cells, we counted the total number of neurons in the measured area and considered these as 100%. Three replicate measurements were performed for each capture region. The number of IR neurons was measured by ImageJ software (v. 1.53o 11 January 2022; http://imagej.nih.gov/ij/index.html).

### 4.4. Statistical Analysis

The statistical analysis was performed using Sigma Plot 12 software (Systat Software, Delaware, USA). The values are reported as the M ± SEM. Differences were assessed by one-way ANOVA with a post hoc Bonferroni’s multiple comparison test. *p* values < 0.05 were considered statistically significant.

## 5. Conclusions

Our results indicate that aging is closely related to changes in the VMH neurochemistry. The molecular mechanisms leading to changes of Lin 28 and components of PI3K/AKT/mTOR signaling pathway in the hypothalamus need to be further explored.

## Figures and Tables

**Figure 1 ijms-23-13468-f001:**
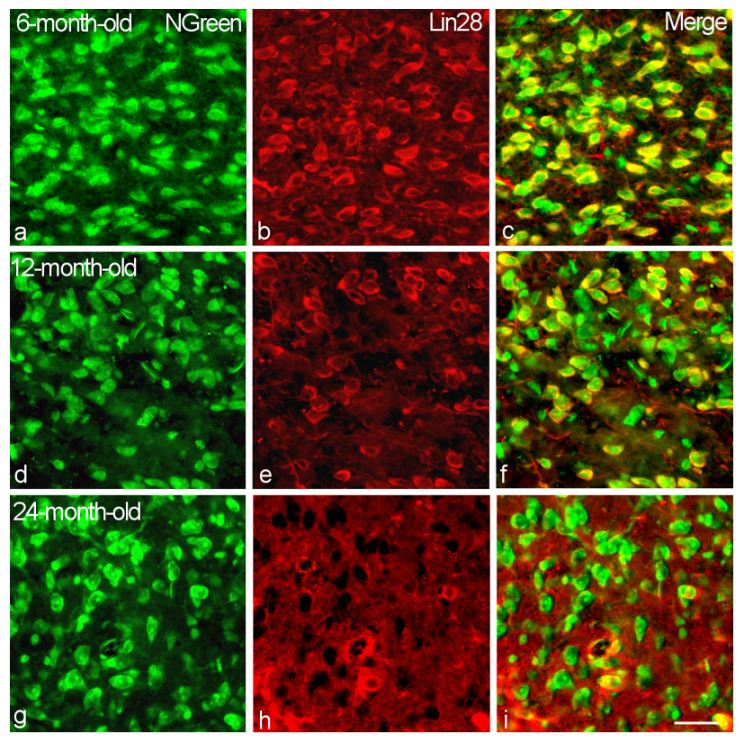
Immunostaining of Lin28-IR neurons in the VMH of 6-month-old (**a**–**c**), 12-month-old (**d**–**f**) and 24-month-old (**g**–**i**) male rats. Double labeling of neuronal marker NGreen (**a**,**d**,**g**, green, left panel), Lin28 (**b**,**e**,**h**, red, middle) and composite image (**c**,**f**,**i**, right). Bar, 25 µm.

**Figure 2 ijms-23-13468-f002:**
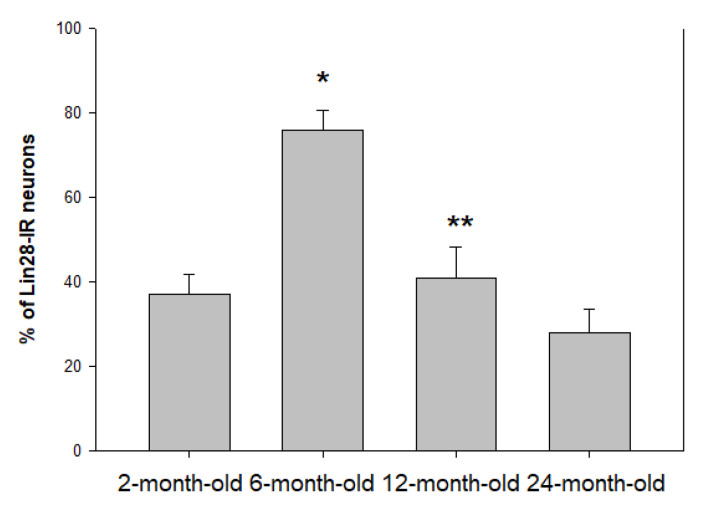
Percentage of Lin28-IR neurons in the VMH of rats of different age group. n = 5 per group (* *p* < 0.01, comparing 2-month-old rats; ** *p* < 0.05, comparing 6-month-old rats).

**Figure 3 ijms-23-13468-f003:**
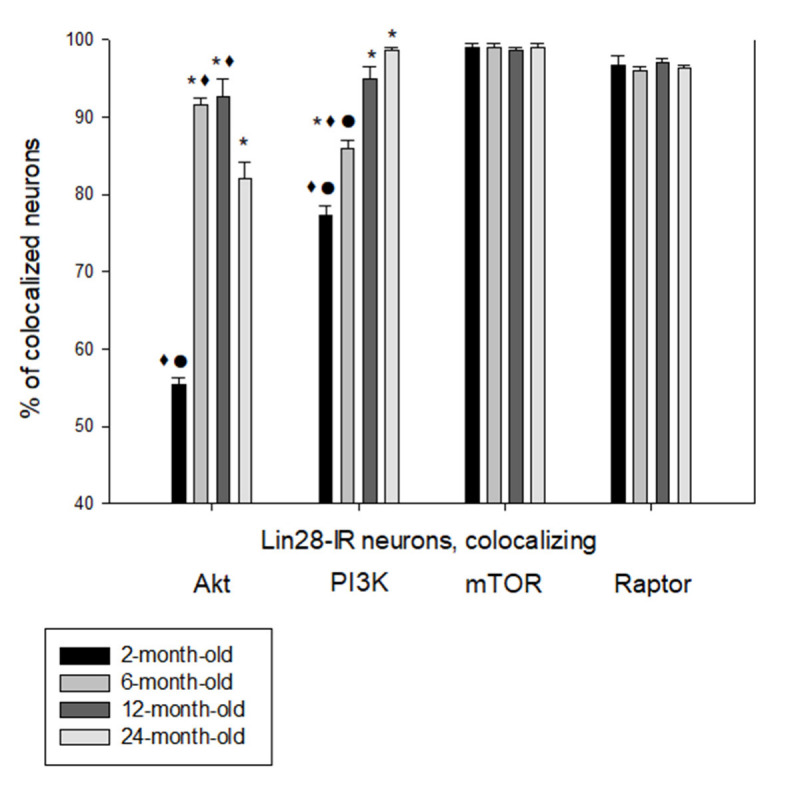
Percentage of Lin28-IR neurons, colocalizing AkT, PI3K, mTOR and Raptor in the VMH of rats of different age group. n = 5 per group (* *p* < 0.05, comparing 2-month-old rats; ● *p* < 0.05, comparing 12-month-old rats; ♦ *p* < 0.05, comparing 24-month-old rats).

**Figure 4 ijms-23-13468-f004:**
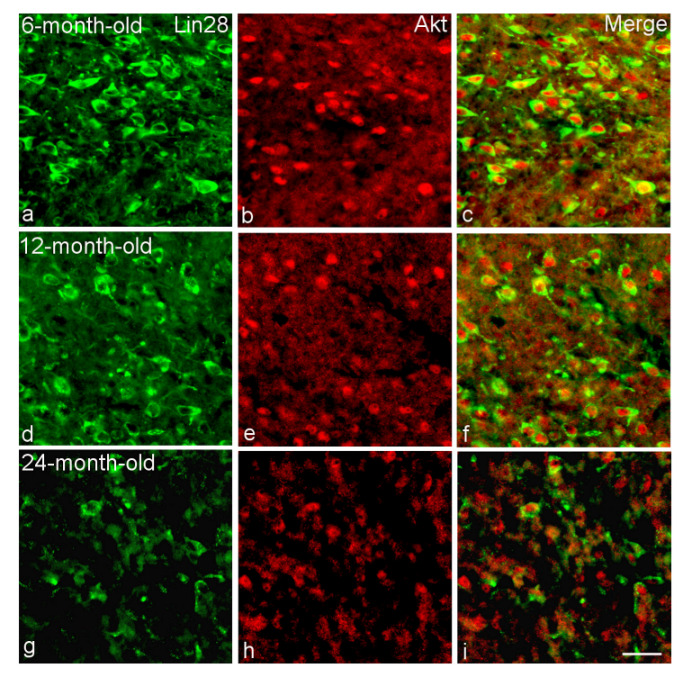
Colocalization of Lin28 with Akt in the VMH neurons of 6-month-old (**a**–**c**), 12-month-old (**d**–**f**) and 24-month-old (**g**–**i**) male rats. Double labeling of Lin28 (**a**,**d**,**g**, green, left panel), Akt (**b**,**e**,**h**, red, middle) and composite image (**c**,**f**,**i**, right). Bar, 25 µm.

**Figure 5 ijms-23-13468-f005:**
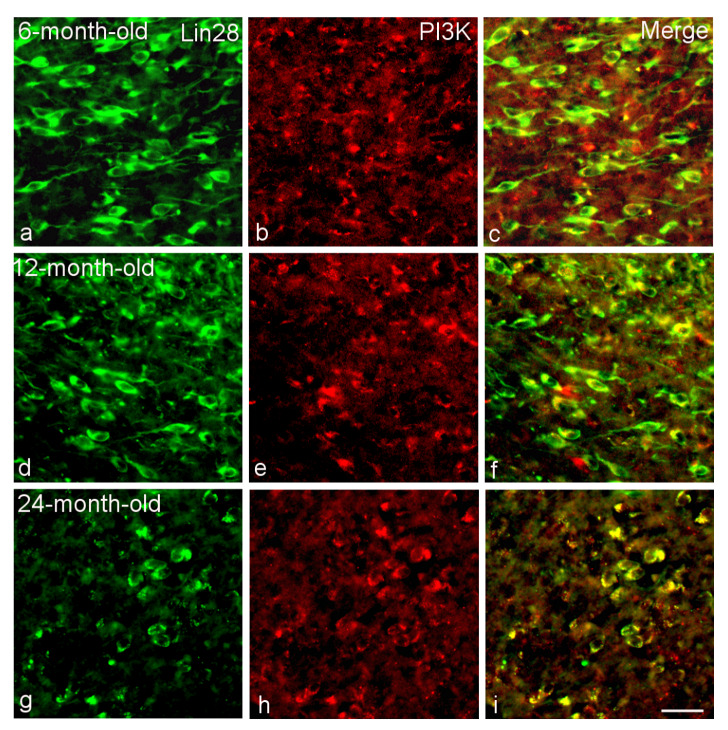
Colocalization of Lin28 with PI3K in the VMH neurons of 6-month-old (**a**–**c**), 12-month-old (**d**–**f**) and 24-month-old (**g**–**i**) male rats. Double labeling of Lin28 (**a**,**d**,**g**, green, left panel), PI3K (**b**,**e**,**h**, red, middle) and composite image (**c**,**f**,**i**, right). Bar, 25 µm.

**Figure 6 ijms-23-13468-f006:**
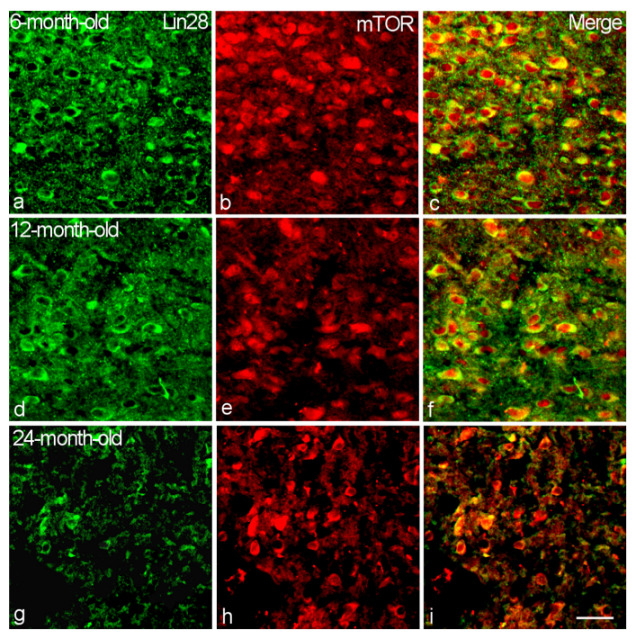
Colocalization of Lin28 with mTOR in the VMH neurons of 6-month-old (**a**–**c**), 12-month-old (**d**–**f**) and 24-month-old (**g**–**i**) male rats. Double labeling of Lin28 (**a**,**d**,**g**, green, left panel), mTOR (**b**,**e**,**h**, red, middle) and composite image (**c**,**f**,**i**, right). Bar, 25 µm.

**Figure 7 ijms-23-13468-f007:**
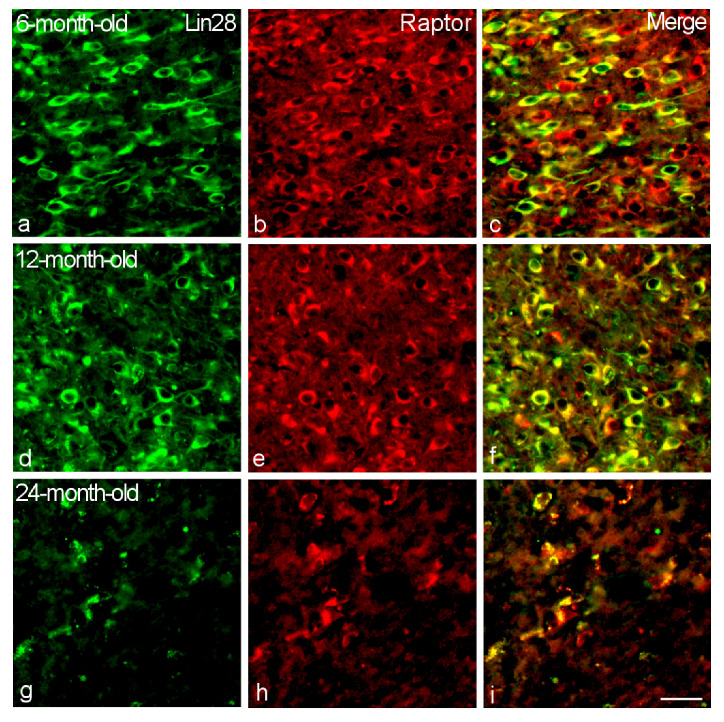
Colocalization of Lin28 with Raptor in the VMH neurons of 6-month-old (**a**–**c**), 12-month-old (**d**–**f**) and 24-month-old (**g**–**i**) male rats. Double labeling of Lin28 (**a**,**d**,**g**, green, left panel), Raptor (**b**,**e**,**h**, red, middle) and composite image (**c**,**f**,**i**, right). Bar, 25 µm.

## Data Availability

The data that support the findings of this study are available from the corresponding author upon reasonable request.

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
