# Peer review of "Ontogenetic Changes in the Expression of the Lin28 Protein in the Rat Hypothalamic Tuberal Nuclei"

_ijms, 2022, doi:10.3390/ijms232113468_

Round 1

Reviewer 1 Report

Review on the manuscript  submitted to International Journal of Molecular Sciences.

The brain hypothalamus is a regulator of homeostasis, biological rhythms, and adaptation to environmental factors, and is also involved in aging. The LIN28 gene encodes an RNA-binding protein containing a cold shock domain at the N-terminus and two retroviral-type CCHC zinc finger motifs at the C-terminus. LIN28 is a marker of undifferentiated embryonic stem cells that binds to the Let7 microRNA precursor and blocks the formation of mature Let7 microRNAs. The authors used immunohistochemistry as a method to examine tuberal hypothalamic neurons expressing the Lin28 gene in male rats at 2, 6, 12, and 24 months. They showed its expression in the ventromedial nucleus (VMH) and no expression in the dorsomedial and arcuate hypothalamic nuclei. The authors also showed a change in Lin28 expression associated with age and its relationship with insulin-related pathways. The work is original, performed using modern technologies, presents interest to the scientific community, and can be published in the International Journal of Molecular Sciences.

Major comments:

  • none

Minor comments:

In the introduction, the authors should describe in more detail the known functions of the Lin28 gene, its role in the insulin signaling pathway, stem cell function, carcinogenesis, and its difference from the Lin28B gene.   

Author Response

Thank you very much for your opinion about the manuscript and your valuable notes. Corrections in the text are highlighted in yellow. Please find below my reply to your comments:

We added additional data about the functions of the Lin28 gene in the beginning of the introduction:

The highly conserved RNA-binding protein, Lin28, is involved in many biological processes, including development, control of onset of puberty and menopause, cell reprogramming, pluripotency, and metabolism. Lin28 was initially discovered in Caenorhabditis elegans as a controller of time development, and it has been found to selectively re-press the expression of microRNAs, including those related to the Let-7 family [1, 2]. Mammals produce two Lin28 paralogs, Lin28 (also known as Lin28a) and Lin28b. Lin28 and Lin28b are differentially localized in cells with predominantly cytoplasmic Lin28, whereas Lin28b accumulates in the nucleus [3].

The Lin28/Let-7 axis is characterized by double negative feedback in the regulation of various biological functions. The level of let-7 is directly correlated with the expression levels of Lin28/Lin28b. Lin28 binds to the terminal loops of let-7 precursors, leading to inhibition of processing and the induction of uridylation and degradation of the precursor. Lin28 is also a direct translational regulator: it selectively binds to a group of mRNAs and stimulates their translation [4, 5].

Lin28 and let-7 regulates the self-renewal and differentiation of stem cells. Lin28 or Lin28b is upregulated whereas let-7a is downregulated in cancer and undifferentiated human and mouse embryonic stem cells relative to normal tissues. The let-7 miRNA family members act as tumor suppressors by inhibiting expression of oncogenes and pluripotency factors including K-Ras, Cyclin D1, c-Myc, Cdc34, Hmga2, E2f2, and Lin28 [6, 7].

Reviewer 2 Report

Expression of Lin28 in the rat hypothalamic tuberal nuclei with aging

Polina A. Anfimova, Lydia G. Pankrasheva, Konstantin Yu Moiseev, Elizaveta S. Shirina, Valentina V. Porseva, Petr M. Masliukov

The article presents the results of a study of the distribution and expression of Lin28 in the neurons of the tuberal nucleus of the hypothalamus and studied its changes depending on age. The colocalization of Lin28-immunoreactive neurons with insulin signaling components, such as mTOR, Raptor, PI3K, and Akt, in the hypothalamic structures was also studied. The Lin28 protein is involved in the implementation of the central regulation of carbohydrate metabolism, and, therefore, this protein may be associated with the development of the metabolic syndrome and the aging process. Such a study was carried out by the authors for the first time, and this, undoubtedly, is the merit of the work. The article is fully consistent with the profile of the journal and is in good agreement with the subject of the Special Issue.

The article was made at a high methodological level, using modern immunohistochemical methods, the data are presented logically, their detailed discussion is carried out. However, there are questions and comments, including on the section "Materials and Methods" (listed below). Regarding the interpretation of the results of the study, there is a remark that consists in insufficient substantiation of the fact that the observed changes in the expression of the Lin28 are functionally associated with aging, and not with a change in the stages of ontogeny.

The major comments

1.The first and main remark is related to the fact that the results described on page 2 indicate that the amount of the investigated Lin28 protein only changed significantly between the 2-, 6- and 12-month groups, and no significant differences were shown for the 24-month group. Based on these data, can we conclude that Lin28 expression is associated with aging? (can a 12-month-old rat be considered an aging animal?). According to the data presented, it can be judged that the studied indicator changes with age and reaches a constant level by the middle of life. Maybe it's better to focus in the title not on the aging process, but on ontogenetic changes in the expression of the Lin28 protein in the rat hypothalamic tuberal nuclei?

2.A similar problem occurs when interpreting the results in paragraph 2.2, which describes colocalization in neurons of the studied protein Lin28 with various components of hypothalamic insulin signaling. For the Lin28/PI3K pair (not sure if it is correct to speak of such a pair), it was shown that the expression of these proteins in the age groups of 6, 12, and 24 months is higher than that in the age group of 2 months, but does not differ within these groups. That is, the revealed changes occur in the hypothalamus of rats between two and six months, but according to these data, such changes cannot, strictly speaking, be attributed to age-related changes in the period of old age. Again, the question of the validity of the title of the article (see the comment 1).

3.There are questions to the Fig. 2 on the Lin28/Akt1 pair. In the text in lines 75-76 it is written that the number of such neurons is maximum in groups of 6 and 12 months compared to groups of 2 and 24 months, while in the figure there is no notation for the significance of differences between the 24 month group and the 6 or 12 month groups. Either they are not statistically significant, and this should be mentioned separately in the text, or the differences are statistically significant, and then this should be indicated in Fig. 2.

4.In the section "Materials and Methods" in the paragraph where the experimental groups are described, the number of animals in each of them is not indicated. This needs to be done.

5. In the section "Materials and Methods" there is no detailed description of the statistical analysis of the obtained experimental data and the criteria by which the groups were compared are not indicated. This needs to be done.

6.In Figure 2, the columns of the diagram rest against the upper border of the graph, and the marker of significant differences in one place goes beyond the border of the figure. To make the graph more correct, you should increase the scale along the y-axis and cut off part of the axis at the base so that the tops of the columns move to the center of the figure.

7.From the Introduction, it is not very clear the role and relevance of studying Lin28 in the nuclei of the hypothalamus, and the peculiarity of representatives of the Let-7 mRNA family. It is necessary to focus on this issue.

8.In the description of the results, it is necessary to add a graph with the percentage of Lin28-IR neurons in VMH (quantitative analysis).

9.Do the authors have data on colocalization of insulin/IGF-1 receptors and Lin28 protein? Otherwise, it is difficult to talk about the relationship with insulin signaling (in the hypothalamus, PI3K and Akt are downstream components of many signaling pathways).

10.Was there a study of the metabolic state in rats of the studied ages, including assessment of glucose and insulin levels, insulin resistance index, and glucose tolerance? If such studies are not done, how valid are the assumptions about the relationship of aging, metabolic syndrome and Lin28 expression (lines 107-111)?

The minor comments

1.In some places, the designations of the experimental groups differ in the text. So, in line 76, the groups are called “1 year” and “2 years”, and in the rest of the text these same groups are called 12 and 24 months, respectively.

2.In the discussion on line 102, lin-28 should be capitalized L. Line 120 - Lin28a appears, nothing was said about this molecule earlier, a brief description is needed.

3.Line 32 - Let 7, in other places of the text Let-7. Capitalized Let-7 in the introduction, lowercase Let-7 in the discussion. Must be presented uniformly. 

Author Response

Thank you very much for your careful revision of the manuscript and suggestions on how to improve the quality of the article. Corrections in the text are highlighted in yellow. Please find below my reply to your comments:

The major comments

1.The first and main remark is related to the fact that the results described on page 2 indicate that the amount of the investigated Lin28 protein only changed significantly between the 2-, 6- and 12-month groups, and no significant differences were shown for the 24-month group. Based on these data, can we conclude that Lin28 expression is associated with aging? (can a 12-month-old rat be considered an aging animal?). According to the data presented, it can be judged that the studied indicator changes with age and reaches a constant level by the middle of life. Maybe it's better to focus in the title not on the aging process, but on ontogenetic changes in the expression of the Lin28 protein in the rat hypothalamic tuberal nuclei?

Thank you very much for your valuable comment. The title has been corrected to: Ontogenetic changes in the expression of the Lin28 protein in the rat hypothalamic tuberal nuclei.

2.A similar problem occurs when interpreting the results in paragraph 2.2, which describes colocalization in neurons of the studied protein Lin28 with various components of hypothalamic insulin signaling. For the Lin28/PI3K pair (not sure if it is correct to speak of such a pair), it was shown that the expression of these proteins in the age groups of 6, 12, and 24 months is higher than that in the age group of 2 months, but does not differ within these groups. That is, the revealed changes occur in the hypothalamus of rats between two and six months, but according to these data, such changes cannot, strictly speaking, be attributed to age-related changes in the period of old age. Again, the question of the validity of the title of the article (see the comment 1).

Statistically significant differences were also found between 6-month-old and 12-month-old rats. Differences between 12-month-old and 24-month-old rats were not statistically significant. Corrections were made in the text.

3.There are questions to the Fig. 2 on the Lin28/Akt1 pair. In the text in lines 75-76 it is written that the number of such neurons is maximum in groups of 6 and 12 months compared to groups of 2 and 24 months, while in the figure there is no notation for the significance of differences between the 24 month group and the 6 or 12 month groups. Either they are not statistically significant, and this should be mentioned separately in the text, or the differences are statistically significant, and then this should be indicated in Fig. 2.

Differences between the 24 month group and the 6 or 12 month groups were statistically significant (p<0.05). Corrections were made in the text.

4.In the section "Materials and Methods" in the paragraph where the experimental groups are described, the number of animals in each of them is not indicated. This needs to be done.

5 animals in each group, 20 total.

  1. In the section "Materials and Methods" there is no detailed description of the statistical analysis of the obtained experimental data and the criteria by which the groups were compared are not indicated. This needs to be done.

We are sorry we missed this important section. Description of the statistical analysis was added into the Methods section.

6.In Figure 2, the columns of the diagram rest against the upper border of the graph, and the marker of significant differences in one place goes beyond the border of the figure. To make the graph more correct, you should increase the scale along the y-axis and cut off part of the axis at the base so that the tops of the columns move to the center of the figure.

Figure 2 was corrected. Since a new graph with the percentage of Lin28-IR neurons was added, fig. 2 has been renamed to fig 3.

7.From the Introduction, it is not very clear the role and relevance of studying Lin28 in the nuclei of the hypothalamus, and the peculiarity of representatives of the Let-7 mRNA family. It is necessary to focus on this issue.

We have added additional data about function of Lin28 in the Introduction. Also, Lin28 is highly expressed in the hypothalamus in contrast with peripheral tissues. Expression Lin28 in the hypothalamus is affected by the metabolic state. Lin28a overexpression in the hypothalamus induced a significant improvement in the glucose metabolism did not influence body weight. The hypothalamus is a primary controller of homeostasis, biological rhythms and adaptation to different environment factors. It also participates in the aging regulation.

8.In the description of the results, it is necessary to add a graph with the percentage of Lin28-IR neurons in VMH (quantitative analysis).

We have added a new graph with the percentage of Lin28-IR neurons in VMH (fig. 2).

9.Do the authors have data on colocalization of insulin/IGF-1 receptors and Lin28 protein? Otherwise, it is difficult to talk about the relationship with insulin signaling (in the hypothalamus, PI3K and Akt are downstream components of many signaling pathways).

Unfortunately, we did not have data on colocalization of insulin/IGF-1 receptors and Lin28 protein. We agree that PI3K and Akt are downstream components of many signaling pathways. In this case, we corrected “insulin signaling” to “PI3K/AKT/mTOR pathway” in the text.

10.Was there a study of the metabolic state in rats of the studied ages, including assessment of glucose and insulin levels, insulin resistance index, and glucose tolerance? If such studies are not done, how valid are the assumptions about the relationship of aging, metabolic syndrome and Lin28 expression (lines 107-111)?

We agree that assumptions about the relationship of aging, metabolic syndrome and Lin28 expression seems to be premature. We removed this paragraph from discussion

The minor comments

1.In some places, the designations of the experimental groups differ in the text. So, in line 76, the groups are called “1 year” and “2 years”, and in the rest of the text these same groups are called 12 and 24 months, respectively.

Corrected

2.In the discussion on line 102, lin-28 should be capitalized L. Line 120 - Lin28a appears, nothing was said about this molecule earlier, a brief description is needed.

Corrected to Lin28.

3.Line 32 - Let 7, in other places of the text Let-7. Capitalized Let-7 in the introduction, lowercase Let-7 in the discussion. Must be presented uniformly. 

Corrected as Let-7.

Round 2

Reviewer 2 Report

The authors have largely revised the article in accordance with the remarks and comments, and they have fully taken into account all 10 main remarks.

The important thing is that they changed the title of the article (comments 1, 2 and 10). The new title "Ontogenetic changes in the expression of the Lin28 protein in the rat hypothalamic tuberal nuclei" is more consistent with the content and conclusions of the article.

The authors introduced information about the statistical analysis of the results obtained, clarified a number of questions regarding the significance of differences between the experimental groups and showed this in the figures. As a result, many of the ambiguities that were in the original version of the article have now been eliminated.

Information about Lin28 and Lin28a is presented in the introduction and further in other sections, which is very important for understanding the studies carried out and the results obtained, as well as for substantiating the purpose of the study (comment 7).

The relationship between insulin signaling, Lin28 and PI3K/AKT is described more cautiously, since PI3K and AKT kinase may be targets for more than just insulin (comment 9).

The authors also checked the text and corrected terminological and syntactical inaccuracies.

Thus, all comments were taken into account, which, of course, further increased the value and scientific significance of the article and the new results obtained.